# Identifying Drug-Therapy Problems among Syrian Refugees in Zaatari Refugee Camp

**DOI:** 10.3390/ijerph19127199

**Published:** 2022-06-12

**Authors:** Alaa M. Hammad, Walid Al-Qerem, Fawaz Alasmari, Jonathan Ling, Raghda Qarqaz, Hakam Alaqabani

**Affiliations:** 1Department of Pharmacy, Faculty of Pharmacy, Al-Zaytoonah University of Jordan, Amman 11733, Jordan; waleed.qirim@zuj.edu.jo (W.A.-Q.); raghdaqar@gmail.com (R.Q.); hakamaqabani@gmail.com (H.A.); 2Department of Pharmacology and Toxicology, College of Pharmacy, King Saud University, Riyadh 12372, Saudi Arabia; ffalasmari@ksu.edu.sa; 3Faculty of Health Sciences and Wellbeing, University of Sunderland, Chester Road, Sunderland SR1 3SD, UK; jonathan.ling@sunderland.ac.uk

**Keywords:** safety, effectiveness, DTPs, drug-therapy problems, Syrian refugees, Zaatari refugee camp

## Abstract

**Background:** Due to a lack of proper pharmaceutical care, Syrian refugees in the Zaatari refugee camp are more likely to have drug-related issues, such as prescription errors and adverse drug occurrences. **Aim:** The current study aims to identify drug-therapy problems among Syrian refugees in the Zaatari refugee camp. **Method:** This is a retrospective cross-sectional study. Patients’ files were collected from the Zaatari camp database. Patients who were 18 years or older and were previously diagnosed with a chronic disease were included. A classification of drug therapy problems (DTPs) was adapted. **Results:** The data of 1530 adult patients (896 females) were collected. The mean age of the sample was 53.7 years and the mean Body mass Index (BMI) was 27.20. The mean of all taken medications was 4.01 (±2.33) medications, with a maximum number of 13. A total of 3572 DTPs was identified, with a mean of 2.33 (±1.26) DTPs per patient. Based on the above-mentioned classification, 70.32% of the DTPs were related to indication, 26.65% were related to effectiveness, and 3.03% were related to safety. **Conclusion:** This study found that refugees in the Zaatari refugee camp have numerous DTPs among their medications. Greater focus should be placed on their medical care, in order to prevent any future complications due to DTPs.

## 1. Introduction

The National Coordinating Council for Medication Error Reporting and Prevention (NCC-MERP) defines a drug error as any preventable incident that may cause or lead to inappropriate medication use or patient harm, while using medications under the guidance of a healthcare provider, patient, or consumer [1]. These occurrences could be linked to numerous factors, including professional practice, healthcare products, compounding, dispensing, patient education, and use [2]. To report and identify medication errors, several classifications have been utilized (for review please see [3]). In this article, medication errors are classified using the Cipolle et al. classification, which uses the term “drug-therapy problem” (DTP) rather than “drug-related problem” [4]. According to the World Health Organization (WHO), drug-therapy problems are defined as “any noxious and unexpected reaction to a medication that happens at doses normally used in man for prophylaxis, diagnosis, or treatments of illness, or for adjustment of physiological function” [5]. Indeed, any undesirable incidence related to medication therapy that actually or potentially impairs planned therapeutic goals is referred to as a DTP [4]. A DTP can happen at any point during the treatment process, although it is most common during the prescribing, transcribing, dispensing, and patient usage of drug therapy [6]. There are four different types of DTPs, including indications, effectiveness, safety, and convenience. These catagories can be further divided into seven different types, including ineffective drug therapy, inappropriate drug therapy, dosages that are too low or too high, adverse drug reactions (ADRs), and non-compliance [6].

DTPs have been linked to poor clinical and financial outcomes in a number of international-health-care systems, including the Netherlands [7], Denmark [8], New Zealand [9], Qatar [10], and Saudi Arabia [11]. In Jordan, a study of the DTPs of hospitalized patients in one hospital’s internal medicine department found a mean DTP rate of 9.4% [12]. Another study looked at DTPs in individuals with chronic conditions, who went to community pharmacies, and found that the average DTP rate was 4.1% [13]. Importantly, a large-scale study conducted in Jordan on a sample of outpatients with chronic diseases displayed an average of 11.2 DTP per patient [14].

Syrians have been fleeing to neighboring countries since the start of the Syrian conflict in 2011. Jordan, which is located south of Syria, receives Syrian refugees on a daily basis. The number of Syrian refugees given official assistance has surpassed 1.4 million (statistics given in 2015) [15]. The Zaatari refugee camp is the world’s largest camp, with an area of 2.0 sq miles (5.2 km^2^) for Syrian refugees, hosting around 81,000 refugees. It has one emergency clinic that can transfer to the nearest govermental hospital and eight healthcare centers. There are 58 full-time clinicians and 150 community-health volunteers, funded by the United Nations High Commission for Refugees (UNHCR). These volunteers are made up of 90% Jordanians and 10% internationals (Qatar, Saudi Arabia, Morocco, United Arab Emirates) and are not restricted to working in the Zaatari refugee camp only, so may have other commitments. The physical wellbeing of refugees is a medical challenge. Due to competition for basic-living necessities as well as a lack of pharmaceuticals and appropriate health care, refugees have a higher frequency of associated DTPs than non-displaced people [16,17]. Due to their impact on mental health and normal human functioning, chronic diseases, whether diagnosed prior to migrating or newly diagnosed after migrating, require specific attention and monitoring [18]. Pharmacists and medications in sufficient numbers are required to control chronic conditions [19]. In a recent study conducted in the Shatila refugee camp in Lebanon, Médecins Sans Frontière’s model of care and treatment outcomes for diabetes and hypertension patients was found to be viable, with positive results among those who had been recruited [20]. Thus, the aim of this study is to identify drug-therapy problems among Syrian refugees in the Zaatari refugee camp, in order to resolve and prevent them in future as well as to provide the best pharmaceutical-care outcomes.

## 2. Method

This is a retrospective cross-sectional study. The data were collected from patients’ files in the Zaatari refugee camp. The collected data included demographics, chronic diseases, medication history, and laboratory tests. Ethical approval was obtained from the Al-Zaytoonah ethical committee, approval number 21/1/2021-2022 on 21 January 2021.

Cipolle et al.’s drug therapy for drugtherapy problems (DTPs) was adapted. Cipolle et al. classified DTPs into four main categories: indication, effectiveness, safety, and compliance [21].

The DTPs were also classified as actual and potential DTPs. Actual DTPs included the DTPs that are confirmed based on medical guidelines. Problems with effectiveness were discovered by comparing patients’ treatment to the most recent clinical-practice evidence-based guideline recommendations. Doses were compared to evidence-based guideline recommendations or drug-information sources such as Lexicomp’s *Drug Information Handbook*, to ensure that the dosage regimen was appropriate [22]. When deciding on the suitability of a dose regimen, clinical characteristics of the patients were considered. Adverse drug reactions were discovered through a review of symptoms and an investigation of patients’ medical records for any probable adverse drug reactions. Potential DTPs included DTPs that require more information to be confirmed. This classification system was used previously in various studies [12,13] to identify the DTPs, which either actually or potentially interfered with the clinical outcomes for each patient. This system was used in the current study to identify unnecessary drug therapy, untreated conditions, ineffective/incomplete drug therapy, inappropriate dosage regimen, adverse drug effects, and actual or potential drug interactions. The atherosclerotic cardiovascular disease risk (ASCVD) score was calculated to estimate the 10-year risk of cardiovascular diseases [23]. The CHA_2_DS_2_-VASc (congestive heart failure, hypertension, age ≥ 75 years, diabetes mellitus, stroke or transient ischemic attack (TIA), vascular disease, age 65 to 74 years, sex category) score was calculated to predict thromboembolic risk in atrial-fibrillation patients [24]. The number of medications was determined by summing the number of all medications that patients take including vitamins and supplements.

### Statistical Analysis

All data analyses were conducted using IBM^®^ SPSS^®^ software version 27, Chicago, IL, USA: SPSS Inc. Continuous variables are presented as means (±standard deviations), and categorical variables are presented as frequencies (percentages). The actual DTPs were categorized into two groups: less than 3 DTPs and 3 or more DTPs. Stepwise binary regression was conducted to determine the variables associated with the number of DTPs. The model included the following independent variables: sex, age, and number of medications.

## 3. Results

Data from 1530 adult patients (896 females) were collected. The demographics and health status of the included patients are presented in Table 1. The mean age of the sample was 53.7 years and the mean BMI was 27.20. One-quarter of the patients were smokers (25.9%). The most common chronic diseases among the patients were hypertension (HTN) (66.5%) and type 2 diabetes mellitus (T2DM) (41.6%).

Table 2 shows the frequency of chronic medications taken by patients. The most-taken medication was simvastatin (58%), followed by aspirin (43%). The mean of all taken medications was 4.01 (±2.33), with a maximum number of 13.

### 3.1. Actual DTPs

The total number of DTPs identified among the current study was 3572, with a mean of 2.33 (±1.26) DTPs per patient. Based on the Cipolle et al. classification, 69.98% of the DTPs were related to indication, 26.98% were related to effectiveness, and 3.04% were related to safety.

### 3.2. Indication

#### 3.2.1. Uncontrolled Diabetes

A cut-off point of 7% HbA1c is generally used to classify patients into controlled and uncontrolled diabetes [25]. Based on this cut-off point, 623 (90.16%) patients in the current study had uncontrolled diabetes (54 patients with uncontrolled type 1 diabetes (98.2%) and 569 patients with uncontrolled type 2 diabetes (90.46%)).

#### 3.2.2. Uncontrolled Hypertension

According to the American College of Cardiology (ACC), the goal of pharmacologic therapy in patients with hypertension is to achieve blood pressure less than 130/80 mmHg [26]. Using this threshold, 987 (97.1%) patients had uncontrolled hypertension.

#### 3.2.3. Unnecessary Drug Therapy

According to the ADA, metformin is not indicated for the treatment of type 1 diabetes [25]; in this study, we have 28 (1.83%) patients diagnosed with type 1 diabetes who are talking metformin as part of their treatment regimen.

#### 3.2.4. Drug Use without Indication

Twenty patients (1.31%) who were taking metformin were found to have no indication for the treatment.

### 3.3. Statins

Based on the American College of Cardiology/American Heart Association Guidelines (ACC/AHA), 631 patients fulfilled the criteria for high-intensity statins. Of these, 235 patients had a preexisting atherosclerotic cardiovascular disease (clinical ASCVD), and 396 patients were 45–75 years of age with diabetes and Low-density lipoprotein cholesterol LDL-C 70–189 mm/dL without clinical ASCVD, but they had an estimated 10-year risk of 7.5% or more [27]. Among those patients, 26 were not taking any type of statins.

Regarding the ACC/AHA criteria for moderate-intensity statins, 143 patients fulfilled these criteria. Six patients ages 40 to 75 years with diabetes and LDL-C 70 to 189 mg/dL without clinical ASCVD, but they had a 10-year ASCVD risk of less than 7.5% or higher. There were 137 patients without clinical ASCVD, 40 to 75 years of age, with LDL-C 70 to 189 mg/dL, who had an estimated 10-year ASCVD risk of 7.5% or higher [27].

#### 3.3.1. Angiotensin-Converting Enzyme Inhibitor (ACEI) or Angiotensin Receptor Blockers (ARBs) for Patients with History of Myocardial Infarction (MI) or Acute Coronary Syndrome (ACS)

According to the ACC/AHA and the American College of Emergency Physicians, all patients with a history of MI or ACS who also have comorbidities should be prescribed ACEI unless contraindicated, and angiotensin receptor blockers (ARBs) should be prescribed for patients who are intolerant to ACEI [28,29]. In this study, 44 patients were eligible for ACEI, based on the aforementioned criteria, but were not prescribed ACEI or ARBs.

#### 3.3.2. Reliever for Asthmatic Patients

Two of the asthmatic patients in this study were not prescribed any type of as-needed reliever.

#### 3.3.3. Aspirin for Secondary Prevention of Cardiovascular Diseases

The National Institute for Health and Care Excellence (NICE) recommends lifelong use of aspirin for secondary prevention in patients with a history of ACS [30] and stable angina [31]. In the current study, 239 patients had this condition, yet 21 patients were not taking aspirin.

#### 3.3.4. Aspirin for Primary Prevention of Cardiovascular Diseases for Diabetic Patients

According to the American Diabetes Association (ADA), aspirin is recommended for primary prevention for patients with DM who are at increased risk of cardiovascular disease (ASCVD ≥ 10) [32]. Among the participants who fulfilled these criteria, 194 out of 395 patients were not taking aspirin.

#### 3.3.5. ACEI and ARBs for Patients with Chronic Kidney Disease (CKD) and Hypertension (HTN)

ACEI and ARBS have a nephroprotective effect for patients with CKD and HTN [33,34]. Sixty-eight participants had CKD and HTN but only forty-four of them were taking ACEI or ARBs.

#### 3.3.6. ACEI and ARBs for Patients with Heart Failure

According to the NICE guidelines from 2018, ACEI or ARBs are considered as a first-line treatment for chronic management for patients with heart failure (HF) and reduced-ejection fraction [35]. Eight patients in the current study had this condition, two of whom were not taking ACEI or ARBs.

#### 3.3.7. Short-Acting Nitrates for Patients with Stable Angina

Short-acting nitrates should be prescribed for patients with stable angina, in order to be used as a reliever treatment during acute angina attacks or for prophylaxis [36]. In this study, 12 patients had stable angina, and none of them were prescribed short-acting nitrates.

#### 3.3.8. Rate-Control Medications for Patients with Atrial Fibrillation (Afb)

Six out of forty-four Afb patients were not treated with beta blockers or calcium-channel blockers for rate control.

#### 3.3.9. Anticoagulants for Patients with Afb

Anticoagulants should be offered to patients with Afb and a CHA_2_DS_2_-VASc score of 2 points or more [37]. Nine males and thirteen females satisfied these conditions, but eight males and none of the females were taking anticoagulants. Moreover, five males had a score of 1 point due to having HTN. Further investigation of these patients’ conditions should be conducted in order to judge if they are candidates for anticoagulation.

#### 3.3.10. Inhaled Corticosteroids (ICS) for Chronic Obstructive Pulmonary Disease (COPD)

In the current study, 30 participants had chronic obstructive pulmonary disease (COPD) and 4 of them were not taking any treatment for COPD. Several studies have found that mono therapy with inhaled corticosteroids (ICS) for patients with COPD does not modify either the mortality rate or the declining of forced expiratory volume in the first second (FEV1), so the usage of short-acting beta agonist (SABA) in combination with ICS alone is not recommended [38]. Among COPD patients in the current study, 1 patient was taking ICS alone and 18 patients were taking ICS with SABA.

### 3.4. Effectiveness

#### 3.4.1. Statins

Among the patients who fulfilled the criteria for high-intensity statins [27], 605 patients were taking low-moderate intensity statins, and among those who were eligible for moderate-intensity statins [27], 42 patients were taking low-intensity.

#### 3.4.2. Antihypertensive Medications

According to the ACC/AHA Task Force on Clinical Practice Guidelines, for hypertensive patients without any comorbidity, beta blockers are inferior to first-line antihypertensive drugs (thiazide diuretics, ACEI, ARBs, and calcium channel blockera) [39]. Therefore, beta blockers should be used only as second-line agents. In the current study, beta blockers were used as a mono-therapy for 51 patients, and were used without the use of any first-line agents for 206 patients.

#### 3.4.3. Metformin

Metformin is the first choice for T2DM and should be used for all patients unless contraindicated [40]. Among this study’s participants, 32 patients were not taking metformin, and they had no clear contraindication (glomerular filtration rate GFR > 45 mL/min). Of those, 16 patients were taking insulin, and 14 patients were taking a drug from the sulfonylurea class.

#### 3.4.4. As-Needed Low Dose ICS-Formoterol

According to the Global Strategy for Asthma Management and Prevention from 2020 (GINA2020), as-needed low-dose ICS-formoterol is the preferred reliever, as it provides better asthma control and better prevention of exacerbation when compared to as-needed short-acting β agonists (SABA) [41]. Among the 40 asthmatic patients in the current study, 38 patients were taking SABA as a reliever instead of ICS-formoterol.

#### 3.4.5. Rhythm-Control Medications for Patients with Afb and Heart Failure

One patient was suffering from atrial fibrillation and heart failure, which may have been induced by Afb. This case warranted the use of rhythm-control drugs [37], which they were not taking.

### 3.5. Safety

#### 3.5.1. Insulin and Sulfonylurea

Four patients were taking insulin with a drug from the sulfonylurea class. This combination is known to increase the risk of hypoglycemia [42].

#### 3.5.2. Metformin

Metformin is contraindicated in patients with GFR < 30 mL/min due to the risk of lactic acidosis [43]. In the current study, 13 patients had GFR < 30 mL/min and were taking metformin.

#### 3.5.3. High Dose ICS

The use of ICS and long-acting β agonists (LABA) in combination in asthmatic patients is proven to be beneficial in improving morning-peak-expiratory flow and increasing the number of symptom-free days [44]. Moreover, high-dose ICS without LABA is not supported by the GINA recommendations [41]. Among the asthmatic patients in the current study, 30 patients were taking high-dose ICS without LABA.

#### 3.5.4. Drugs Used in Asymptomatic Hyperuricemia

The main potential clinical consequences of hyperuricemia are gout, urate nephropathy, and nephrolithiasis. Several agents should be avoided, when possible, in patients with hyperuricemia, to prevent the promotion of hyperuricemia/incident gout. These agents include thiazide or loop diuretics, ACEIss, non-losartan ARBs, and beta blockers [45]. Among the patients with hyperuricemia in this study, 20 were taking diuretics, 20 were taking ACE inhibitors or ARBs, and 20 were taking beta blockers.

#### 3.5.5. Concomitant Use of Beta Blockers and Beta-2 Agonists

Non-selective beta blockers such as propranolol may antagonize the effects of salbutamol and precipitate acute, life-threatening bronchospasm in patients with asthma or other obstructive-airway diseases [46]; therefore, concomitant use should generally be avoided. In the current study, one patient was taking this combination.

#### 3.5.6. Potential DTPs

The total number of potential DTPs was 738 DTPs and was classified as follows: 266 indication-related DTPs, 7 effectiveness-related DTPs, and 465 safety-related DTPs.

### 3.6. Indication

#### 3.6.1. Beta Blockers for Patients with History of MI or ACS

According to AHA/AACF, beta blockers should be prescribed for patients with normal left-ventricular function who have had MI or ACS continuing for three years. Among the patients who fulfilled this criterion in the current study (although the date of MI/ACS is uncertain), 44 patients were not taking beta blockers.

#### 3.6.2. Dual Antiplatelet for Patients with History of MI or ACS

Different guidelines suggest that the combination of aspirin and P2Y12 inhibitors should be continued for at least one year in patients receiving drug-eluting stents and for up to one year in those receiving bare metal stents [47]. In this study, only 7 out 229 patients were taking dual-antiplatelet therapy.

#### 3.6.3. Undiagnosed Hypertension

Similar to the above threshold, 282 patients were found to have high blood pressure but were not diagnosed in their files as being hypertensive patients.

### 3.7. Effectiveness

#### Short-Acting Bronchodilators

The short-acting muscarinic antagonist (SAMA) mono therapy has small benefits when compared to SABA mono therapy, in the aspects of lung function, health status, and requirements for oral steroids. Moreover, the combination of SABAs and SAMAs provides better results in terms of FEV1 and improvement of symptoms, when compared to either treatment alone [38]. Seven of the COPD patients in this study were taking SABA alone.

### 3.8. Safety

#### 3.8.1. Glucagon-Like Peptide-1 Receptor Agonists for Patients with T2DM

Based on the ADA recommendations [40], Glucagon-like peptide-1 Receptor Agonists (GLP-1 RA) should be considered for most patients who needed injectable therapy prior to the initiation of insulin. Among the type 2 diabetic patients who were taking insulin in the current study (90 patients), none were prescribed GLP-1 RA.

#### 3.8.2. Rhythm-Control Medications for Patients with Afb

None of the atrial-fibrillation patients were opted for rhythm control. However, due to the lack of other variables that elicit the decision of conversion of rate control to rhythm control, further evaluation should be conducted to assess these patients’ eligibility for rhythm-control drugs [37]. Table 3 illustrates different DTPs.

#### 3.8.3. Major Drug–Drug Interactions

Five major drug–drug interactions [48] were noticed among the participants, and these interactions include the concomitant use of the following drugs: enalapril with allopurinol (31 patients), atenolol with verapamil (1 patient), warfarin with aspirin (23 patients), verapamil with simvastatin (4 patients), and amlodipine with simvastatin (273 patients); medications are dispensed depending on the availability within the clinic and the cost of the medication.

## 4. Discussion

This is the first study to look at DTPs among Syrian refugees in the Zaatari refugee camp in Jordan. According to our findings, Syrian refugees had an average of around three DTPs. The most generally identified DTPs were related to indications, safety, and effectiveness. To the authors’ best knowledge, no study focusing on DTPs has been conducted in Syria, and few studies on DTPs have been conducted in Jordan. Indeed, a study of DTPs among hospitalized patients at one hospital’s internal-medicine department discovered a 9.4% DTP rate [9]. A large-scale investigation in Jordan revealed an average of 11.2 DTPs per patient in a group of outpatients with chronic diseases [11]. These studies, therefore, had a higher DTPS incidence than our own findings, with three DTPs per person. Importantly, a similar study to our results examined DTPs in people with chronic illnesses who visited community pharmacies and discovered that the average DTP rate was 4.1% [10]. This rate aligns with the findings of a study conducted in Brazil, which reported a mean of 3.1 ± 1.5 DTPs per patient [49]. A previous study on Syrian refugees with diabetes and hypertension in the Shatila refugee camp, Lebanon, showed that 8% had type 1 diabetes and 30% had type 2 diabetes [20], while another study conducted on Palestinian refugees showed that type 1 diabetes was present in 4.3% and type 2 diabetes in 95.7% [50]. These findings are parallel to those we found, of around 4% with type 1 diabetes and 41.5% with type 2 diabetes. Moreover, in the Shatila refugee camp, 30% had hypertension [20] and 68.5% of Palestinian refugees had hypertension [50], while in our study 66.5% had hypertension. The most-common disease found among out-of-camp Syrian refugees in Jordan was hypertension, followed by arthritis, diabetes, chronic respiratory diseases, and cardiovascular disease [51]. Furthermore, another study in Jordan on outpatients with chronic diseases found that the most common disease was hypertension, followed by diabetes, coronary heart disease, peptic ulcers, and respiratory-related diseases [13]. In our study, the most common diseases were hypertension, followed by diabetes, cardiovascular diseases, and respiratory diseases.

The most common DTPs among Syrian refugees were assigned as “indication”, as indicated by the uncontrolled status of HTN and T2DM. Importantly, additional drug therapy was needed, including adding statins, ACEI, or ARBS for patients with a history of cardiovascular disease, according to the latest guidelines. Moreover, adding a reliever for asthmatic patients, aspirin as a secondary or primary prevention measurement, short-acting nitrates, rate-control medication for Afb patients, and anticoagulants, were needed, plus there was inadequate use of ICS for COPD patients. Parallel to the current results, a previous study showed that drug-related problems among older people with dementia had many DTPs classified as “needs additional drug therapy”, such as patients who were provided opiates without laxatives or who were under-prescribed medicines for heart failure [52].

In the Shatila refugee camp, 39% of refugees who were enrolled in the study had uncontrolled diabetes [20], while in our study around 90% of the enrolled refugees had uncontrolled diabetes; this might be due to the difference in the cut-off point of HbA1c used, which was 8% vs. 7% in this current study. Indeed, a study conducted on Palestinian refugees in Jordan, using a similar cut-off point of HbA1c, found that the percentage of patients with uncontrolled diabetes is higher in patients with type 1 diabetes (92.6%) compared to type 2 diabetes (74.9%) [50], which is parallel to our results of 98.2% and 90.46%, respectively. Furthermore, 50% of Syrian refugees in the Shatila refugee camp who entered the study had uncontrolled hypertension [20]. Our results indicated that we had 97.1% with uncontrolled hypertension, although this might be due to the different goala set for hypertension, of 140/90 vs. 130/80.

Interestingly, 282 (18.2%) patients in the current study were found to have high blood pressure but were not diagnosed as hypertensive. These patients need close monitoring and repeated measurement for hypertension, in order to be diagnosed correctly [26]. This DTP might be due to patient-related barriers, including poor medication adherence, patients’ beliefs about hypertension and its treatment, depression and other cognitive dysfunction, low health literacy, comorbidities, patient motivation, coping, and lack of social support. Moreover, physician-related barriers play an important role, including a lack of intensity of drug therapy (also known as clinical inertia), communication style, and awareness and knowledge of treatment guidelines. Moreover, factors related to the medical environment or healthcare system, including a lack of access to care, cost of medications, and low socioeconomic status, play an important role as well [53]. More studies are warranted to investigate the reason for undiagnosed hypertension.

Another potential DTP of importance among Syrian refugees is labeled as “safety”; including prescribing both insulin and sulfonylurea, which increases the risk of hypoglycemia [42], and dispensing metformin for patients with GFR < 30 mL/min that would increase the risk of lactic acidosis [43]. Moreover, using high-dose ICS without LABA for asthmatic patients, which is not supported by the GINA recommendations [41], and prescribing contraindicated drugs for patients with asymptomatic hyperuricemia, including ACEI as well as concomitant use of beta blockers and beta-2 agonists, was observed. Many factors related to drug therapy must be examined, while conducting a medication review. Changes in pharmacokinetics and pharmacodynamics that are associated with renal and hepatic dysfunction may increase the likelihood of side effects [54].

Gilbert et al. [55] reported that “the need for laboratory testing” is the most common DRP in Australia, but “inappropriate medications” and “the need for additional drug therapy” were the most common DRPs in other studies from the United States and Taiwan [56,57]. Roughead et al. [58] observed that the most common DTPs were “further monitoring”, “wrong drug”, “more medicine is needed”, and “using too little of the drug”. The most common DTPs in Minnesota, according to Rao et al. [59], were “further drug is required”, “using too little of the drug”, and “non-compliance”. Non-adherence issues connected with self-medication and adverse drug responses were the most common medication-usage-related problems among outpatients in a Nigerian study [60]. Moreover, studies conducted in Jordan found that the major DTPs are “more effective therapy available” [13], and “a need for additional or more frequent monitoring” [14]. The discrepancy that is found among different studies conducted in different countries may be explained due to the differences in patient age, lifestyle, disease prevalence, type of drugs, and number of drugs. Comparisons between pieces of research are also challenging due to variances in methodologies and situations. In our study, medication reviews were undertaken by researchers (academic clinical pharmacists), but in others, community pharmacists in conjunction with general practitioners [17,20] or community pharmacists in partnership with geriatricians [19] were involved. A Jordanian study reported similar results to ours, where a comprehensive assessment of treatment-related problems in hospitalized medicine patients and main DTPs were “indication-related problems” [12].

According to the current study, the indication DTPs percentage was the highest (69.96%). This differs from previous studies, which reported that indication DTPs in hospitalized medicine patients in Jordan were 16.44% [12]. Emergency-department visits and admissions due to drug-related problems in Saudi Arabia were 12.5% [11], and the drug-related problems in a sample of outpatients with chronic diseases were 10.47% [14]. Our study showed that effectiveness DTPs frequency was 26.97%. This aligns with a study conducted on hospitalized medicine patients in Jordan (30.66%) [12]. These frequencies are higher than drug-related problems in emergency-department visits and admissions due to drug-related problems in Saudi Arabia (12.5%) [11], and drug-related problems in a sample of outpatients with chronic diseases (5.59%) [14]. Moreover, safety frequency was the lowest in a study that investigated the drug-related problems of outpatients with chronic diseases in Jordan (1.23%) [14], followed by this current study (3.07%). A study of drug-related problems in emergency-department visits and admissions due to drug-related problems in Saudi Arabia and the current study, which assessed the treatment-related problems in hospitalized medicine patients in Jordan, showed a much higher safety frequency (37.5% and 24.97%, respectively) [11,12]. This discrepancy in the DTPs reported might be due to the difficult circumstances that the refugees are living in and the overwhelming of the health-care sector in the Zaatari refugee camp, alongside the number of patients who were enrolled in the study.

In the current study, 1017 participants had hypertension, and 625 of them used Enalapril (61.5%), followed by 417 that used amlodipine (46.3%). This is similar to a previous study conducted in Hiwot Fana Specialized University Hospital in eastern Ethiopia, in which Enalapril was the most frequently used drug followed by calcium-channel blockers [61]. In another study conducted in Indonesia, calcium-channel blockers (85.09%) and ACEI (42.48%) were the most frequently used medications for hypertension [62]. Moreover, a similar study conducted in Malaysia found that the highest-used anti-hypertensive medications were calcium-channel blockers and ACEI [63].

In this study, 629 participants have type 2 diabetes, and597 of them used metformin (94.9%), followed by glimepiride 204 (32.43%). Importantly, 55 participants have type 1 diabetes, and 28 of them take metformin (50.91%); according to the American Diabetes Association (ADA), metformin is not indicated in type 1 diabetes [64]. Moreover, 20 patients were taking metformin with no indication. In addition, a study in Malaysia found that insulin and metformin were the most frequently used medications for type 2 diabetes [63]. A similar study in eastern Ethiopia showed that metformin and insulin account for 37.2% and 16.2% of frequently used medications for type 2 diabetes, respectively [65].

In this research, simvastatin was the most frequently used medication among participants for dyslipidemia (58.37%), followed by gemfibrozil (0.07%), both as treatment and preventive drugs. Nevertheless, in a previous study, simvastatin was used at a higher rate (72.6%) when compared to the current study, followed by rosuvastatin or pravastatin (0.5%) [66]. Most of the participants in this research took lower doses instead of higher or moderate doses of statins (83.6%), compared to another study conducted in Malaysia, which showed a dosing problem, when the drug dose is too low or the dosage regimen is not frequent enough (10.1%) [66]. However, many participants in this study did not take statins at all (4.13%), which was lower than the previous study conducted in Malaysia (11.3%) [66].

In this study, DTPs were more common among patients who were prescribed a larger number of medicines. The likelihood of DTPs was linked to patients with a larger number of medicines, which has been previously reported [52,67,68]. The number of medicines taken increases exponentially rather than linearly, with the number of drugs consumed [69]. Patients taking four or more medicines, for example, are much more likely to have ADR-related hospital admissions than those prescribed three or less prescriptions (11.1% vs. 3.6%) [70]. Importantly, being prescribed a larger number of drugs has also been connected with an increased risk of hospitalization, partially due to drug–drug interactions [71]. Indeed, another study found that hospitalized patients who had an ADR took twice as many medicines (12.5 vs. 6.3) as individuals who did not have an ADR [72].

The current findings revealed a link between the presence of DTPs and a variety of medical disorders, including hypertension and diabetes. This is in line with the findings of previous studies, which revealed that the drugs used to treat illnesses (e.g., diuretics, digoxin, insulin, and oral antidiabetics) were associated with DTPs as risk factors among patients [61]. Indeed, a previous study reported that patients with a psychiatric diagnosis had high DTPs (69.6%) [73].

Despite having access to healthcare, refugees continue to be afflicted by diseases as well as DTPs [74]. The health issues that refugees experience is reflective of the general health conditions and trends in Syria and Jordan as well as, to some extent, the rest of the world. The disease epidemiology among Syrian refugees is similar to that of many other countries throughout the world. Syrian refugees suffer from chronic ailments, communicable infections, injuries, and mental and emotional disorders [75]. Moreover, Syria was reported to have a high burden of noncommunicable diseases, even before the commencement of the war [51]. Refugees’ health has deteriorated as a result of their displacement from their homes and loss of resources.

Despite the fact that the risk factors for many chronic diseases are mostly manageable, refugees are more concerned with and focused on surviving than on controlling their chronic disorders or DTPs [74]. Simultaneously, their refugee status makes them more vulnerable to a variety of ailments, both chronic and acute as well as both communicable and noncommunicable. As a result, health-promotion and illness-prevention services should be enhanced, and acute-care hospital and health-center services should be upgraded and expanded [51]. Changes in the Jordanian Ministry of Health’s healthcare policy on refugee access to health services in 2014 obliged refugees to bear the price of medicines and consultations, which has had a significant impact on them [76].

A key strength of this study is the large sample size, which includes almost all the Syrian refugees living in the Zaatari refugee camp, who have a record of having a chronic disease. However, this study has some weaknesses, such as only describing the current state of DTPs without implementing actual interventions, assessing the acceptance rate of other health care providers to these interventions, and assessing the extent of intervention implementation as well as its impact on patient health outcomes.

## 5. Conclusions

In conclusion, this research has revealed the significant problems with DTPs experienced by refugees living in the Zaatari refugee camp. More studies are warranted to address these problems. Importantly, urgent work is needed to improve the safe and optimal use of medicines in this challenging and resource-poor environment.

## Figures and Tables

**Table 1 ijerph-19-07199-t001:** Sample demographics and health status.

	Frequency (%) or Mean (±SD) (*n* = 1530)
Age	53.7 (±14.00)
Sex	
Female	896 (±58.60)
Male	634 (±41.40)
Body mass index	27.20 (±3.97)
Smoking status	
Smoker	397 (±25.90)
Non-smoker	1133 (±74.10)
Participant health status	
Asthma	40 (±2.60)
Cancer	3 (±0.20)
Chronic obstructive pulmonary disease	30 (±2.00)
Type 1 diabetes mellitus	55 (±3.59)
Type 2 diabetes mellitus	629 (±41.11)
Post-myocardial infarction or acute coronary syndrome	229 (±14.97)
Stable angina	12 (±0.78)
Peripheral arterial disease	26 (±1.70)
Hypertension	1017 (±66.47)
Heart failure	8 (±0.52)
Vitals and laboratory tests	
Systolic blood pressure	137.01 (±8.19)
Diastolic blood pressure	90.37 (±6.57)
Fasting blood glucose	156.58 (±19.07)
Aspartate aminotransferase	38.26 (±12.91)
Alanine transaminase	42.55 (±13.80)
Hemoglobin A1c%	8.08 (±1.64)
Glomerular filtration rate	73.86 (±17.76)
Uric acid	336.80 (±104.05)
Triglycerides	180.17 (±26.68)
High-density lipoprotein	34.44 (±3.94)
Low-density lipoprotein	104.38 (±13.67)

**Table 2 ijerph-19-07199-t002:** Chronic medications.

Medication	Frequency (%)
Aspirin	659 (43.07)
Allopurinol	66 (4.31)
Amlodipine	483 (31.57)
Atenolol	129 (8.43)
Beclomethasone_250-inhaler	63 (4.12)
Bisoprolol	419 (27.39)
Candesartan	12 (0.78)
Carvedilol	1 (0.07)
Clopidogrel	14 (0.92)
Enalapril	686 (44.84)
Furosemide	181 (11.83)
Gemfibrozil	1 (0.07)
Glibenclamide	177 (11.57)
Gliclazide	2 (0.13)
Glimepride	204 (13.33)
Hydrochlorothizaide	179 (11.70)
Insulin	136 (8.89)
Metformin	645 (42.16)
Methyldopa	9 (0.59)
Propranolol	33 (2.16)
Salbutamol	79 (5.16)
Simvastatin	893 (58.37)
Verapamil	5 (0.33)
Warfarin	48 (3.14)

**Table 3 ijerph-19-07199-t003:** Drug-therapy problems (DTPs).

	Actual DTPs		Potential DTPs	
	Number of DTP	Percent of DTP	Number of DTP	Percent of DTP
Indications	2512	70.32%	548	53.74%
Effectiveness	952	26.65%	7	0.67%
Safety	108	3.03%	465	45.59%
Total number	3572	100%	1020	100%

## Data Availability

Not applicable.

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
