# Peer review of "Identifying Drug-Therapy Problems among Syrian Refugees in Zaatari Refugee Camp"

_ijerph, 2022, doi:10.3390/ijerph19127199_

Round 1

Reviewer 1 Report

The manuscript can be accepted now in its present form.

Reviewer 2 Report

The authors have revised their original manuscript according to the reviewers’ comments. I would think that this revised manuscript is better organized and suitable for publication.

This manuscript is a resubmission of an earlier submission. The following is a list of the peer review reports and author responses from that submission.

Round 1

Reviewer 1 Report

This is a very good manuscript dealing with a very important topic identifying drug therapy errors amongst syrian refugees.

I recommend publishing the manuscript after aswering the following questions:

1- How can the authors explain the high number of cases that were prescribed amlodipine + simvastatin, specifically (273 drug-drug interactions)?

2-  Regarding the anti-hypertensive agents: Was the 51 patients who received beta-blockers as a monotherapy without the use of all first-line agents due to illetracy of the guidelines or due to lack of availability?

Reviewer 2 Report

This is a well written and conceived study that documents drug therapy problems in a Syrian refugee camp located within Jordan. The major deficiency with the manuscript is lack of both intra and extra-study context as mentioned below. The authors need to address this well to significanly improve the manuscript. There is little if any additional data to collect or analyses to be made in my opinion which should make it easier for the authors.

Major comments

  1. “In this article, medication errors are classified using the Cipolle et.al classification, which uses the term ”drug-therapy problems” (DTP).” Please reference. Why this specific classification rather than others? Pls justify
  2. The background of the identified problem is rather limited in the manuscript and does not highlight any previous similar studies in other parts of the world. This would help correlate the findings with existing data. A quick literature search shows a few similar studies. E.g.  A study focusing on the two most common condition in another Syrian refugee camp in Lebanon was published in 2019. (Confl Health 2019 Apr 2;13:12)
  3. The setting of the study needs more details. How many refugees are in the camp, how big is the camp, how many doctors, nurses etc.; where do these healthcare workers come from? Are they volunteers? Funding? Are they only providing a service to the clinic or do they have other commitments This would create a scene for the workload and expertise of the staff and it is well established that many of these factors impact on the subject matter.
  4. “Actual DTPs included the DTPs that are confirmed based on medical guidelines. Potential DTPs included DTPs 64 that require more information to be confirmed.” This needs elaboration. Medical guidelines is too vague.
  5. A head to head brief comparison between the population in the camp and a similar population e.g. Jordanians or pre-war Syria would put these findings in context. The details of individual problems is excellent but the context in a broader global population, similar population not in a camp and other refugee camps is lacking and this is a major shortfall.

Minor comments

  1. “To report and identify medication errors, several classifications have been utilized.” Please reference
  2. “The number of Syrian refugees given official assistance has surpassed 1.4 million[4].” Please state the last date on which this data was applicable.
  3. “CHA2DS2‑VASc score” please write in full at first use
  4. Table 1 there are too many abbreviations and the shorter ones can be written in full
  5. Beclomethasone_250 – specify inhaler

Reviewer 3 Report

  I would think this is a valuable report that investigates the current situation and problems of drug therapy in the Syrian refugee camp. Although the data is valuable, I think it does not reach a certain level as a scientific paper due to insufficient comparisons with data in previous reports. A more thorough search of literature, detailed analysis of the differences, and a more in-depth discussion about the differences between the data obtained in this study and the existing previous reports are needed to add.